# Use of Uncrewed Aerial System (UAS)-Based Crop Features to Perform Growth Analysis of Energy Cane Genotypes

**DOI:** 10.3390/plants14050654

**Published:** 2025-02-21

**Authors:** Ittipon Khuimphukhieo, Lei Zhao, Benjamin Ghansah, Jose L. Landivar Scott, Oscar Fernandez-Montero, Jorge A. da Silva, Jamie L. Foster, Hua Li, Mahendra Bhandari

**Affiliations:** 1Texas A&M AgriLife Research and Extension Center, Weslaco, TX 78596, USA; ittipon.kh@tamu.edu (I.K.); jorge.dasilva@ag.tamu.edu (J.A.d.S.); 2Department of Plant Production Technology, Faculty of Agricultural Technology, Kalasin University, Kalasin 46000, Thailand; 3Texas A&M AgriLife Research and Extension Center, Corpus Christi, TX 78406, USA; lei.zhao@ag.tamu.edu (L.Z.); benjamin.ghansah@ag.tamu.edu (B.G.); jose.landivarscott@ag.tamu.edu (J.L.L.S.); osc.fernandezmontero@ag.tamu.edu (O.F.-M.);; 4College of Engineering and Computer Science, Texas A&M University-Corpus Christi, Corpus Christi, TX 78406, USA; 5Texas A&M AgriLife Research-Beeville Station, 3507 U.S. 59 E., Beeville, TX 78102, USA; 6Department of Mechanical and Industrial Engineering, Texas A&M University-Kingsville, Kingsville, TX 78363, USA; hua.li@tamuk.edu

**Keywords:** uncrewed aerial systems (UAS), biomass, plant breeding, logistic function

## Abstract

Plant growth analysis provides insight regarding the variation behind yield differences in tested genotypes for plant breeders, but adopting this application solely for traditional plant phenotyping remains challenging. Here, we propose a procedure of using uncrewed aerial systems (UAS) to obtain successive phenotype data for growth analysis. The objectives of this study were to obtain high-temporal UAS-based phenotype data for growth analysis and investigate the correlation between the UAS-based phenotype and biomass yield. Seven different energy cane genotypes were grown in a random complete block design with four replications. Twenty-six UAS flight missions were flown throughout the growing season, and canopy cover (CC) and canopy height (CH) measurements were extracted. A five-parameter logistic (5PL) function was fitted through these temporal measurements of CC and CH. The first- and second-order derivatives of this function were calculated to obtain several growth parameters, which were then used to assess the growth of different genotypes with respect to weed competitiveness and biomass yield traits. The results show that CC and CH growth rates significantly differed among genotypes. TH16-16 was outstanding for its ground cover growth; therefore, it was identified as a weed-competitive genotype. Furthermore, TH16-22 had a higher CH maximum growth rate per day, yielding a higher biomass compared to other genotypes. The CH-based multi-temporal data as well as the growth parameters had a better relationship with biomass yield. This study highlights the application of UAS-based high-throughput phenotyping (HTP), along with growth analysis, for assisting plant breeders in decision-making.

## 1. Introduction

Global warming is the greatest threat that humankind is currently facing. Among pollutants, carbon dioxide is the main culprit due to the very high levels of anthropogenic emissions released from fossil fuel utilization [1]. In addition to environmental concerns, the rising global demand for energy and depletion of fossil fuel supplies have raised interest in renewable energy such as bioenergy and biofuels [2]. Perennial C_4_ crops, for example, energy sorghum (*Sorghum bicolor*), sugarcane, and energy canes (*Saccharum* spp.), are considered promising feedstock crops for the South Central and Southern U.S. regions because the favorable weather conditions would allow maximum biomass production [3].

Energy cane, a hybrid of sugarcane (*Saccharum* Spp.) and its wild type (*Saccharum spontanerum* L.) is selected for its high biomass instead of high sucrose content [4]. It has emerged as a potential lignocellulosic crop for second-generation ethanol production due to its high biomass yield and adaptation. Furthermore, it is expected to play a significant role as biomass feedstock for bioenergy and bio-based products. Moreover, the dedicated energy cane can also be grown in nutrient-poor marginal lands, which are not suitable for food-crop production; hence, energy cane as a bioethanol production crop will not replace existing food-crop land [5]. Because of global concern regarding the energy crisis and greenhouse gas emissions, research regarding energy cane in many aspects has been carried out, such as agronomic performance [6] and optimum cultivating practice [4]. Despite such progress, this crop is relatively new. Therefore, there are still unexplored topics which need to be investigated to maximize crop productivity. Plant breeding, a genetic improvement, is one of the very first steps that need to be implemented to achieve this goal. The phenotype is a result of genetics, the environment, and their interaction, while its measuring process is called phenotyping. Plant growth analysis is used to describe a biological process where plant characterization is required. In plant breeding, growth analysis can assist breeders in decision-making, because it can explain the morphological, physiological, and environmental reasons behind yield variation among genotypes [7]. At the organ or plant scale, growth is often characterized as tissue expansion over short time periods, while over longer periods, the increase in biomass is a common metric [8]. Thus, plant growth analysis at plant or plot levels usually requires measuring biomass or plant area at successive time points.

Destructive sampling procedures are usually carried out to determine aboveground dry matter for growth analysis; nevertheless, this method is labor-intensive and time-consuming, especially in growth analysis, because it requires high-temporal data. Moreover, plant breeding trials usually utilize an experimental design replicated three to six times, but high-temporal destructive sampling may affect the final yield assessment. Increasing the number of replications to avoid the effect of destructive biomass sampling on final yield assessment is one of the solutions, but this approach significantly increases the resources necessary for yield trials. As a result, there have been attempts to estimate aboveground biomass in sugarcane using its yield components such as the number of stalks, stalk diameter, and plant height [9]. Still, these manual assessments could be labor-intensive and time-consuming, especially in plant breeding trials.

Uncrewed aerial system (UAS)-based high-throughput phenotyping (HTP) makes it possible to obtain high-temporal phenotype data in a nondestructive manner. Canopy cover (CC) and canopy height (CH), which are the result of plant growth, can be measured using a sensor mounted on a UAS. This HTP method is time-efficient; for example, Tanger et al. [10] reported that the height of rice (*Oryza sativa* L.) was manually measured at a rate of 45 plots h^−1^, compared to 3000 plots h^−1^ when measured using HTP. In addition to its time efficiency, it was found to be highly accurate in sugarcane with R^2^ values of 0.89 and 0.81 [11,12] when compared with manual measurements. Similarly, UAS-based CC has been found to be a yield-related feature in sugarcane and energy cane [3,13].

Plant growth generally follows a sigmoidal growth function, which is characterized by an early lag, exponential growth, and a stationary phase [7]. Among several functions for growth modeling, the logistic model is the most used form [14,15], and there are multiple versions of the logistic model. Simple models often fail to capture the complexity or nuances of data, although they require relatively few observations for training [16]. The five-parameter logistic (5PL) function is an extension of the four-parameter (4PL) function, where the fifth parameter is included in the 4PL function, allowing asymmetry to be effectively modeled [17].

Even though the concept of using logistic functions to explain plant growth has been widely reported [15,16,18], leveraging UAS-based phenotyping as input data to overcome the labor-intensive nature of traditional phenotyping for growth analysis is relatively new. Therefore, the objectives of this study were to investigate the potential of using growth functions to obtain growth parameters for UAS-based CC and CH measurements in energy cane. The study further assesses the growth of several energy cane genotypes using the derived growth parameters. Additionally, we also examined the relationship of UAS-based plant phenotypes and growth parameters with dry biomass at different growth stages and the fresh yield of energy cane.

## 2. Results

### 2.1. Temporal UAS-Based Canopy Cover and Canopy Height

The first round of UAS data collection was conducted on the 21st day after planting (DAP), and a significant difference among the seven genotypes for CC was observed (Figure 1a,b). However, there was no significant difference in the later flights at the sprouting stage (34–57 DAP). TH16-16 appeared to be the fastest ground-covering genotype at the sprouting stage (Figure 1a,c), whereas TH16-24 showed the opposite trend. Furthermore, CC was significantly different among the genotypes in all data collected during the tillering stage, except for 64 DAP. As in the sprouting stage, TH16-16 was still outstanding, while TH16-24 was inferior at this point. Once it entered the elongation stage, the cane suffered from an extremely low temperature in February 2024, causing the leaf to be burnt. This resulted in CC dropping to almost zero. Interestingly, Ho02-113 had the fastest recovery, as it had a greater CC than the other genotypes at 166, 176, and 187 DAP. The recovery of CC took roughly 40 days to reach its previous maximum. There were significant differences in CC among the genotypes in all data collected during the elongation stage. The CC during the elongation stage was steady with slight fluctuation, ranging from 70 to 85%. At the early maturity stage (301 DAP), the CC of TCP10-4928 significantly dropped below 60%, whereas the others remained steady up to 351 DAP, but a significant drop was observed at 361 DAP due to leaf senescence.

For CH at the sprouting stage, there was no significant difference for the studied genotypes; in addition, significant growth in this stage was not found (Figure 2a,b). Nevertheless, significant increases in CH measurements were detected during the tillering stage. In comparison with CC, CH took a longer period to recover from cold burn, as the CH was still lower than its previous maximum at 70 days after the incident. Like in the case of CC, Ho02-113 was the fastest-recovering genotype, as it had the highest CH at 187, 207, 218, and 224 DAP. Subsequently, CH growth reached its maximum point around 361 DAP; afterward, a significant decline was seen at 382 DAP due to lodging. The elite lines TH16-22, TH16-16, TH16-18, and TH16-13 outperformed the check cultivar (Ho02-113) according to the heatmap analysis; in addition, TH16-22 was, by far, the tallest genotype, followed by TH16-16, whereas THC10-4928 alongside TH16-24 was the shortest one.

Highly comparable results of variance components and repeatability were observed for both CC and CH. For CC, during sprouting to the early tillering stage (21–78 DAP), there was low variance in genotype components, but the variance in replication and the random error were high; similarly, repeatability during this stage was also low. Overall, an acceptable genotype variance and repeatability for selection was observed during the mid-tillering stage, all the way to the maturity stage. However, the genotype variance and repeatability significantly dropped at the last two dates of data collection (382 and 392 DAP), possibly due to varying canopy senescence across all genotypes and replications. As in CC, high error and low repeatability were observed for CH during the establishment stage (21–78 DAP). Acceptable genotype variance and repeatability were detected at the tillering stage, starting from 84 DAP to 135 DAP, but these did not remain the same after the cane suffered from cold temperatures. The repeatability of CH slowly declined once the cane entered the maturity stage, and it significantly dropped at the last two flights, like that of CC. (See Figure 3).

### 2.2. Five-Parameter Logistic Growth Function Analysis

#### 2.2.1. Canopy Cover-Based Growth Analysis

The CC-based growth analysis across energy cane genotypes is shown in Figure 4. Briefly, the fitted 5PL function allows us to estimate growth at a particular time. The first order derivative describes the rate of change of the function (also known as the growth rate), and the second order derivative describes how the growth rate changes (acceleration or deceleration). Overall, the CC data obtained from the sprouting to the maturity stages of energy cane fitted well with the 5PL function. The change in early growth (T1) was observed at only 2 DAP, and the maximum growth rate (T2) was observed at 63 DAP with a rate of 0.89% day^−1^ (Figure 4b). After the inflection point, the acceleration rate was negative, indicating the deceleration of growth (Figure 4c). Afterward, the growth rate decreased constantly and approached the upper asymptote (T3) at 147 DAP.

The canopy cover growth parameters derived from the 5PL function are shown in Table 1. There was no significant difference in duration 1 (D1) among the seven energy cane genotypes. Nevertheless, D2, D3, and the maximum growth rates were significantly different among the genotypes studied. TH16-24 had the lowest duration for both D2 and D3; in contrast, Ho02-113 showed the opposite trend. TH16-16, by far, had the highest maximum growth rate, with a rate of 1.19% day^−1^; on the other hand, TH16-24 and TH16-22 showed the opposite trend.

#### 2.2.2. Canopy Height-Based Growth Analysis

When all data collected throughout the growing season were fitted with the 5PL function, CH growth did not fit the function, possibly due to discontinued growth for up to 3 months, which was caused by the extremely cold temperature in February. Therefore, the CH growth analysis was divided into two stages: before (BC) and after cold burn (AC).

Before cold burn

In terms of the comparison between CC and CH, a rising curve of CH was observed 3 days later than that of CC. The maximum growth rate was observed at 69 DAP with a rate of 1.91 cm day^−1^ (Figure 5a,b). Afterward, a plateau was observed at 137 DAP, but this saturation was due to the interruption of the severe cold event. For CH growth parameters, significant differences among the genotypes for D1 and the maximum growth rate were observed (Table 2). In addition to TH16-13 having the shortest duration for D1 with only 50.75 days, its maximum growth rate was outstanding as well. However, TH16-22 was, by far, the vertically fastest-growing genotype with a rate of 2.26 cm day^−1^; on the other hand, the growth rate of TCP10-4928 was the slowest.

After a cold burn

After the cane suffered from cold temperatures, CH growth followed the 5PL growth function with a lag phase, exponential growth, and saturation (Figure 6). A comparison between before and after severe cold burn showed that the acceleration rate of CH growth after severe cold burn was not as exponential as it previously was (Figure 5c and Figure 6c); consequently, the maximum growth rate was slightly lower with a rate of 1.51 cm day^−1^, which was observed at 279 DAP (Figure 6b). A non-significant difference among the genotypes for CH growth parameters after cold burn was observed for all growth parameters (Table 3).

#### 2.2.3. Weed-Competitive Genotype Identification

The 5PL function was used to estimate how many days each genotype requires to cover 25%, 50%, and 70% of the plot area by canopy (Figure 7a). The genotypes did not show significant differences in occupying 25%, but significant differences were observed for 50% and 70%. As expected, TH16-16 took a significantly shorter time to occupy 50% and 70%, followed by TH16-18. In contrast, TH16-24 appeared to be the slowest-growing genotype, where it took approximately 3 and 7 weeks longer to reach 50% and 70%, respectively, compared to TH16-16. Furthermore, the vertical growth of TH16-16 was also vigorous.

### 2.3. Dry Biomass, Fresh Yield, and Their Relationship with Temporal Canopy Features and Growth Parameters

#### 2.3.1. Analysis of Variance and Mean Comparison

At the tillering stage (93 DAP), there was a significant difference in biomass among the genotypes, where TH16-22 had the highest dry biomass, followed by TH16-13. In contrast, TH16-18 yielded the lowest biomass at this stage (Figure 8). At the elongation stage (218 DAP), TH16-22 was still outstanding, with an average of 2.19 ± 0.88 kg, followed by Ho02-113. Conversely, TH16-24, with an average of 0.78 ± 0.28 kg, was the lowest-yielding genotype. However, biomass collected at 305 DAP in the maturity stage did not show a significant difference among genotypes, in which high random errors were observed at this stage (Figure 8). Similarly, non-significant differences were observed among the genotypes for fresh yield. Taking dry biomass at all growth stages and fresh yield into consideration, TH16-22 was, by far, the best-yielding genotype, followed by Ho02-113. In contrast, TCP10-4928 and TH16-16 were opposite.

#### 2.3.2. Association Between Temporal Canopy Features and Growth Parameters with Biomass and Yield

Correlation of temporal canopy features and dry biomass at different growth stages

The relationship of dry biomass with temporal CC and CH is shown in Figure 9. Regarding a comparison between CC and CH, the correlation between temporal CH and dry biomass was higher than that between dry biomass and CC for all three growth stages. In comparisons among the growth stages, the highest associations were detected at the tillering stage for both CC (R^2^ = 0.56; *p* < 0.01) and CH (R^2^ = 0.63; *p* < 0.01), followed by the elongation stage. In contrast, neither CC nor CH was significantly correlated with dry biomass at the maturity stage (305 DAP) (Figure 9c).

Correlation of temporal canopy features and growth parameters with fresh yield

The correlation between twenty-six temporal CC and CH measurements and the fresh yield of energy cane is shown in Figure 10. Four temporal CH measurements were significantly associated with the fresh yield of energy cane, and all of these CH measurements (305, 333, 351, and 361 DAP) were collected during the maturity stage. This indicates that the taller genotypes are likely to provide a higher fresh yield than the shorter ones. In contrast, only one temporal CC measurement was significantly correlated with fresh yield, which was found at the early growth stage (34 DAP).

The correlation between canopy growth parameters derived from the 5PL function is shown in Figure 11. The fresh yield of energy cane was significantly correlated with the D1, D3, and maximum growth rate of CH growth parameters before cold burn. The correlation coefficients of D1 and D3 were negative, with coefficients of −0.49 and −0.48, respectively, whereas that of maximum growth rate was positive. Neither CC growth parameters nor CH growth parameters after cold burn were significantly associated with the fresh yield of energy cane.

## 3. Discussion

### 3.1. Temporal UAS-Based Canopy Cover and Canopy Height

The growth and development of *Saccharum* spp. hybrids are divided into four stages, including sprouting (0–60 DAP), tillering (60–150 DAP), elongation (150–240 DAP), and maturity (240 to 360 DAP or more) [19,20,21]. The newly germinated sprouting cane from the seed pieces contributed to CC at the sprouting stage (0–60 DAP). A significant difference in CC at the first flight (21 DAP) indicated that the time taken to sprout from the seed pieces differed among genotypes, which is common in *Saccharum* spp. hybrids [22]. Although the cane seed piece contains all nutrients and water necessary for germination, the gemination process is sensitive to temperature, where varieties differ in their degree of temperature sensitivity [23]. At the end of this stage (60 DAP), CC occupied roughly 30% of the plot area. Afterward, the tillering stage begins, where the newly sprouted plants germinate from the axillary buds at the base of those shoots that are already established. As expected, the CC doubled from 35% to 70%, approximately a month after the cane entered the tillering stage, and it also reached its plateau at this stage (135 DAP). The comparison between CC and CH growth clearly showed that energy cane at the sprouting stage prioritized expanding ground cover instead of vertical development. At this point, the growth of CH was minimal; however, it had a significant increase after the cane entered the tillering stage. At the elongation stage (150–240 DAP), there is an increase in size due to cell elongation, as a process of plant phenology [21]; therefore, it was expected that the exponential growth of CH would be observed at this stage, but the cane encountered an extremely low temperature right after entering the elongation stage, resulting in CH dropping, and the growth of CH was lagging afterward. It took up to 3 months for the CH of the cane to fully recover from the cold burn, since it suffered from such burn during its critical stage of vertical growth (elongation), unlike CC, which was already saturated when the cold burn occurred.

Low genotype variance and repeatability at the sprouting stage for both CC and CH were observed, indicating high environmental effects, which had mostly been due to the varying germination rate from the cane pieces between replications of a particular genotype, leaving some visually observable gaps in the plots. However, genotype variance and repeatability increased significantly after the cane entered the tillering stage, because these gaps were filled with new plants germinated from the established shoots.

### 3.2. Five-Parameter Logistic Growth Function Analysis

The fitted logistic function and the first and second derivatives of the function serve as the basis for constructing growth, growth rate, and growth acceleration curves, respectively [24]. The concept of using UAS-based phenotype data, combined with the growth function, is relatively new. According to our knowledge, Bhandari [7], for the first time, used UAS-based CC alongside several growth functions, and he found that the 4PL function was the best fit. However, he did not use CH growth in his study. For the CC-based growth analysis in this study, large variation at the maturity stage was observed as shown in Figure 4, because it was averaged across seven genotypes, in which the leaf senescence of TCP10-4928 was observed earlier than that of the others. As expected, a maximum CC-based growth rate occurred during the tillering stage at 63 DAP, where TH16-16 had the highest maximum CC growth rate.

Unlike for CC, when CH data collected throughout the growing season were fitted, they did not follow the 5PL function. Our hypothesis for this observation is the discontinuation of vertical development for up to 3 months at the elongation stage, due to cold burn. Nevertheless, CH growth was analyzed separately before and after the cold burn. The results show that these data fitted well with the 5PL function in both periods. TH16-22, before cold burn, overperformed compared to the others, while CH growth after cold burn was not significantly different among the genotypes. At early growth stages (sprouting and tillering), leaf appearance and leaf extension mostly contributed to the growth of CH; in contrast, internode expansion mainly contributed to CH growth at the elongation stage. In general, CC-based growth did not significantly translate to dry biomass yield, because we found that the fastest ground-cover-growing genotype (TH16-16) had low biomass yield; in other words, TH16-22 was identified as a high-yielding genotype, but its ground cover growth was not vigorous. In contrast, CH growth agreed with dry biomass yield, as the vertically fastest-growing genotype (TH16-22) also had the highest biomass yield.

Selecting early vigor and weed-competitive energy cane is a promising approach for sustainable weed control by suppressing weed establishment [25]. This would result in reducing herbicide use, as well as cultivation. This is a very important attribute, especially for energy cane, because it is expected to be grown under low-input conditions. Vigorous and early vegetative growth is one of the general features of a weed-competitive genotype [26,27]. In *saccharum* spp. breeding, breeders consider this feature a valuable trait, but due to the unavailability of an efficient approach for such an assessment, the canopy closure and canopy growth are usually rated using categorical observation with categories such as slow, intermediate, or fast, which limits statistical analysis to differentiate the tested genotypes. One of the promising applications of using UAS-based CC, along with the growth function, is to overcome the above-mentioned challenges. Considering the speed after planting needed to cover 50% and 70% of the plot area as an indicator, TH16-16 and TH16-18 were identified as weed-competitive genotypes. TH16-16 was, by far, the most outstanding one. It took only three months for it to cover 70% of the area, which was two months less than TH16-24. Moreover, its vertical growth was relatively vigorous as well. This approach would allow plant breeders to overcome the challenge of canopy closure phenotyping.

### 3.3. Dry Biomass, Fresh Yield, and Their Relationship with Temporal Canopy Features and Growth Parameters

Although significant differences in the dry biomass of the seven genotypes at the tillering and elongation stages were observed, a similar trend was not observed during the maturity stage due to high random error. Likewise, fresh yield was also not significantly different among the genotypes. This study is consistent with previous reports by Cholula et al. [3] and Knoll et al. [28] who found a highly positive relationship between CH and biomass. It was clear that vertical growth was more correlated with dry biomass than ground cover development for all growth stages. Leaf appearance and expansion mainly contribute to ground cover growth, while the contribution of vertical growth is mainly from stalk elongation. Nevertheless, leaves and stalks are important contributing factors in biomass accumulation. Leaves, which are source tissue where photosynthesis takes place, export over 80% of fixed photosynthates to sink tissue (stalk) at maturity stage [29]. Our results were relatively comparable with those reported by Zhao et al. [30], who found that an increased biomass yield of energy cane is primarily associated with a higher stalk population and longer stalk, rather than the leaf net photosynthetic rate (Pn). After canopy closure (218 and 305 DAP), the correlation between temporal CC and dry biomass was poor, as compared with in the early stage. Our hypothesis for such an observation is that this happens due to canopy saturation. The dry biomass of the energy cane was increasing because of growth, whereas UAS-based CC had already reached its plateau due to the inability to detect new leaves appearing, resulting in the poor relationship between dry biomass and temporal CC at the later stage. Similarly, the correlation between the twenty-six temporal CC measurements and fresh yield was also poor, in which only one temporal CC measurement (34 DAP) was significantly correlated with fresh yield. In contrast, a better association between temporal CH and dry biomass was observed as compared to CC; moreover, a significant correlation between temporal CH and fresh yield was observed for several time points. A possible hypothesis for this observation is that CH, unlike CC, is not affected by canopy closure, resulting in a better relationship with biomass. Furthermore, to obtain a higher correlation between temporal CH and final yield, an optimum time at which to perform UAS data collection during the maturity stage would be recommended. This finding is consistent with a previous report in sugarcane [13].

As found for temporal CC, CC growth parameters derived from the 5PL function were not significantly correlated with the fresh yield of energy cane. Canopy saturation may have been responsible for such an observation. The CC maximum growth rate was found to be significantly correlated with biomass in wheat (*Triticum aestivum* L.) [7]. However, CH growth parameters, including D1, D3, and the CH maximum growth rate, were significantly correlated with fresh yield, which is consistent with the previously mentioned results derived from temporal canopy features and fresh yield. This result suggests the importance of the canopy height of energy cane in accumulating biomass yield.

### 3.4. Limitation and Future Research

The canopy cover was saturated at approximately 130 days after planting, but the energy cane was still in its vegetative growth stage at this time; therefore, the energy cane was still growing after canopy saturation. However, the proposed CC measurement method was not able to detect such growth because of canopy saturation. This is because the approach in this study is not able to penetrate through the canopy. In addition, although the CH derived from Structure from Motion (*SfM*) is not affected by canopy saturation, the CH itself does not significantly reflect the stalk volume. To overcome this challenge, fitting the function with a point cloud derived from a light detection and ranging (LiDAR) sensor would be valuable in future research, not only because this sensor is able to penetrate through canopy, but also because its point cloud is likely to significantly reflect canopy volume. Therefore, fitting high-temporal LiDAR data with a growth function may allow for a better logistic-based growth model in assessing the biomass of energy cane. In addition, another interesting topic is the validation of UAS-based growth, compared to traditional growth analysis (e.g., destructive biomass sampling).

## 4. Materials and Methods

### 4.1. Study Area and Experimental Design

The trial was carried out at Texas A&M AgriLife Research and Extension, Weslaco, Texas (Figure 12). The climate in this area is characterized by hot, humid summers and generally mild to cool winters with rainfall averaging 645.16 mm annually. Before planting, soil samples were collected for soil fertility analysis (Figure 12b). Thirteen subsamples were randomly collected throughout the trial. All subsamples were mixed thoroughly, and then they were divided into three subsamples for analysis. The soil fertility in the trial shown in Table 4 was averaged from the three subsamples. Overall, the soil had moderate alkaline and slight salinity. The experimental design was a randomized complete block design (RCBD) with 7 genotypes including Ho02-113, TCP10-4928, TH16-13, TH16-16, TH16-18, TH16-22, and TH16-24 with 4 replications. Ho02-113 and TCP10-4928 are check cultivars, while the other five genotypes are elite lines, derived from the AgriLife energy cane breeding program. The energy cane was planted manually on 31st August 2023 with 4 rows/plot, and the length of the plots was 10 m/row. The GoField Plus system (Goanna Ag Pty. Ltd., Goondiwindi, QLD, Australia) was used to measure the crop water status in each plot and determine when irrigation needed to be applied. The crop was cultivated to remove weeds and ensure that the trial was weed-free before UAS flights to minimize noise from non-energy cane species. This study involves data from the plant cane stage, starting from August 2023 to December 2024.

### 4.2. Uncrewed Aerial System-Based Phenotyping Pipeline

#### 4.2.1. Image Acquisition

A DJI phantom 4 RTK (SZ DJI Technology Co., Ltd., Shenzhen, China), equipped with a 20-megapixel, one-inch complementary metal–oxide–semiconductor (CMOS) red, green, and blue (RGB) camera, was used to capture the visible spectrum. Eight ground control points (GCPs) were installed and surveyed using Emlid Reach RS2 (Emlid Tech Kft., Budapest, Hungary) to obtain the precise coordinates of each GCP for georeferencing. The flight was carried out at an altitude of 25 m above ground level with 80% overlapping. The images were acquired on a sunny day from 10 AM to 2 PM with wind speeds less than 10 miles per hour to minimize the effect of fluctuating ambient sunlight levels. Twenty-six flight missions were carried out throughout the growing season (Figure 13). In this study, we used an RGB camera, which does not require radiometric calibration [31]. Therefore, the raw RGB images were further processed to construct an orthomosaic without preprocessing.

#### 4.2.2. Image Processing

The raw images were processed using Airsoft Metashape software version 1.7.1 (Agisoft LLC, St. Petersburg, Russia). This software uses Structure from Motion (*SfM*) to reconstruct 3 dimensions, by using the set of overlapping UAS images. The pipeline reported by Bhandari et al. [32] was followed. Furthermore, the coordinates of each GCP were imported during orthomosaic generation, with the aim of improving geometrical accuracy among multi-temporal UAS flights. Orthomosaics were exported as TIFF images for feature extraction.

#### 4.2.3. CC and CH Extraction

QGIS version 3.34.10. was used to extract CC using the Canopeo algorithm, previously presented by Patrignani et al. [33]. This method is an automatic color threshold (ACT) image analysis tool, based on the red-to-green (R/G) and blue-to-green (B/G) color ratios and excess green index (ExG) [34]. In addition to its high-speed and easy procedure, Canopeo has been found to be efficient in various scenarios such as in varying backgrounds beyond soil and field settings [35], less sensitive to sunlight level compared to the light interception method [34], and able to accurately detect all green parts of plants exposed to sunlight, and shaded leaves [33]. The classification of this method is based on Criterion (1):(1)$$\frac{R}{G}<P1\;and\;\frac{R}{G}<P2\;and\;2G-R-B>P3$$
where R, G, and B represent digital number (DN) values for the red, green, and blue bands, respectively. The default parameter values for Canopeo are P1 = 0.95, P2 = 0.95, and P3 = 20 [33]. The canopy height model (CHM) was also extracted using QGIS version 3.34.10. using Formula (2):(2)$$CHM=Digital\;Surface\;Model\;\left(DSM\right)-Digital\;Terrain\;Model\;\left(DTM\right)\;$$
where DTM was obtained prior to planting. The 95th percentile of CH was extracted using ArcGIS Pro version 2.7.0 (Esri, Redlands, CA, USA).

### 4.3. Dry Biomass and Fresh Yield Data Collection

Biomass samples were collected over a one-meter-long area from the ground to the top of the plant. The data were collected at 93, 218, and 305 DAP, where the cane was in the tillering, elongation, and maturity stages, respectively. The biomass samples were air-dried at 65 °C, using a Precision-Quincy vertical flow air dryer, for a period of 14 days. The dried biomass samples were then weighed. Fresh yield (final yield) was harvested at the end of the season using a Cameco 2500 sugarcane harvester and weighed with a small-capacity weigh wagon (3-Ton Weigh Wagon, CAMECO, Saskatoon, SK, Canada). A summary of ground data collection is shown in Table 5.

### 4.4. Data Analysis

Temporal CC and CH were analyzed according to the RGBD design. RStudio version 2024.09.1 was used to perform a heatmap analysis. We used the scale () function in RStudio to standardize the data. Afterward, the package “ComplexHeatmap” in RStudio was used to graphically generate a heatmap. Variance component decomposition was calculated based on Equation (3):(3)$$Y_{ij}=µ+T_{i}+\beta_{j}+{ɛ}_{ij}$$
where $Y_{ij}$ represents the individual observation of genotype *i* at the replication *j*, µ represents the grand mean, $T_{i}$ represents the effect of the *i*-th genotype, $\beta_{j}$ represents the effect of the *j*-th block, and ${ɛ}_{ij}$ represents the error term. Additionally, repeatability was calculated using Equation (4):(4)$$R=\frac{\sigma_{g}^{2}}{\sigma_{g}^{2}+\sigma_{e}^{2}}$$
where *R* represents repeatability, *σ*^2^*_g_* represents genotypic variance, *σ*^2^*_p_* represents phenotypic variance, and *σ*^2^*_e_* represents environmental variance. The logistic growth function was chosen to describe the growth of energy cane in this study, because it is the most used asymptotic function for plant growth [14,15]. There are several versions of logistic functions. The 1PL and 2PL functions do not often fit well with observations [16]. The 3PL function has a fixed lower horizontal asymptote at 0 and the inflection point falls rigidly, while the 4PL function is not affected by these constraints; nevertheless, the 3PL function provides a more parsimonious and equally adequate fit in some situations, as compared to the 4PL function [16]. However, the 3PL and 4PL functions did not consistently fit across all genotypes according to the preliminary data analysis in this study. The 5PL function, on the other hand, provides maximum flexibility [17] and alleviates both restrictions [15]. Our objective was to assess the growth of several energy cane genotypes, so flexibility was required to ensure that the function fits well with observations across all genotypes. Therefore, the 5PL function was chosen and was calculated using Equation (5):(5)$$f\left(x\right)=d+\frac{\left(a-d\right)}{{\left\{1+({\frac{x}{c})}^{b}\right\}}^{g}}$$
where *a* represents the lower asymptote, *d* represents the upper asymptote, *c* represents the inflection point, *b* represents slope, and *g* represents the asymmetry factor. After the growth curve was constructed, time (T) and growth rate were obtained. Three times, including T1, T2, and T3, were defined. T1 is the lower asymptote where the curve starts rising. T2 is inflection where the maximum growth rate occurs, and T3 is the upper asymptote where the curve flattens approaching saturation (Figure 4). Canopy growth parameters were used to describe the crop phenological growth of each energy cane genotype at plot levels, based on UAS-based HTP. Duration 1 (D1) represents the period from the beginning of vegetative growth to the inflection point, D2 represents the period from the inflection point to the saturation phase, and D3 represents the period from the beginning of vegetative growth to the saturation phase. Canopy growth parameters, including D1, D2, and D3, were obtained using Equations (6), (7), and (8), respectively.(6)$$Duration\;1\;\left(D1\right)=Time\;2\;\left(T2\right)-Time\;1\;(T1)$$(7)$$Duration\;2\;\left(D2\right)=Time\;3\;\left(T3\right)-Time\;2\;(T2)$$(8)$$Duration\;3\;\left(D3\right)=Time\;3\;\left(T3\right)-Time\;1\;(T1)$$

The maximum growth rate was obtained from the first derivative of the function. The least significant difference (LSD) was used to perform a mean comparison at the significance level of 0.05%. The coefficient of determination (R^2^) and correlation coefficient (r) of temporal canopy features and canopy growth parameters with biomass and yield were calculated.

## 5. Conclusions

In this study, we leveraged UAS-based HTP in obtaining successive phenotype data of the ground cover and vertical growth of energy cane throughout the growing season, and the aerial phenotype data were fitted with the 5PL function for growth analysis. Furthermore, the correlation between dry biomass and temporal CC and CH at different energy cane growth stages was investigated in addition to fresh yield. Overall, temporal CC and CH fitted with the 5PL function. TH16-16 was found to be the fastest ground-cover-growing genotype; therefore, it was considered a weed-competitive genotype, while TH16-22 was found to be the vertically fastest-growing genotype, in addition to having a high biomass yield. According to relationships with dry biomass, vertical growthcontributed to dry biomass more than ground cover growth for all growth stages. It would not be recommended to use ground cover growth as an indicator in assessing dry biomass after canopy closure; instead, vertical growth would be recommended for such a scenario. Similarly, CH growth parameters derived from the 5PL function were significantly correlated with fresh yield, but this was not detected for CC growth parameters. In addition, only temporal CH obtained during the maturity stage was found to be correlated with fresh yield. This study provides useful information to assist plant breeders in their decision-making.

## Figures and Tables

**Figure 1 plants-14-00654-f001:**
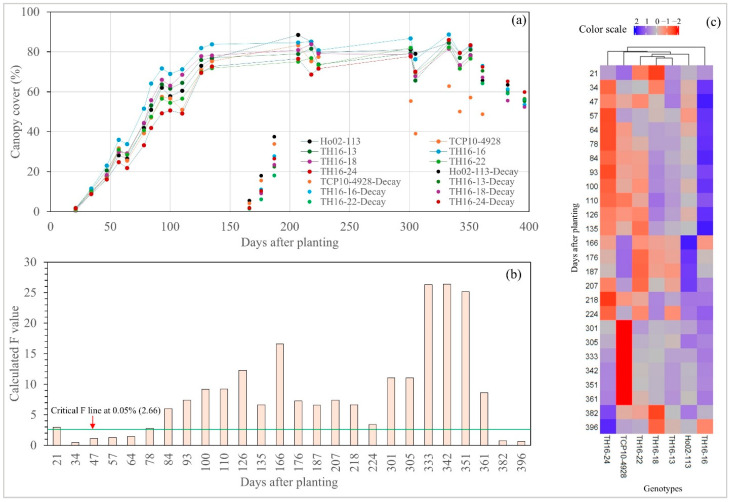
The canopy cover (CC) of seven energy cane genotypes (**a**), a computed F bar graph with critical F value line (**b**), and heatmap analysis based on CC throughout the growing season (**c**).

**Figure 2 plants-14-00654-f002:**
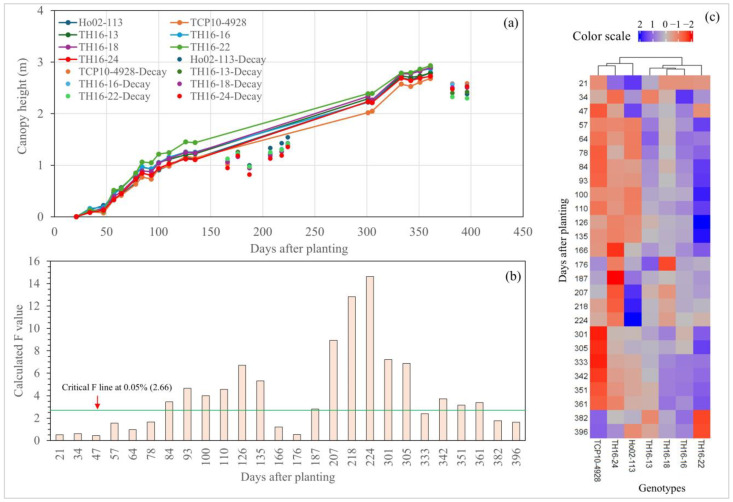
The canopy height (CH) of seven energy cane genotypes (**a**), a computed F bar graph with critical F value line (**b**), and heatmap analysis based on CH throughout the growing season (**c**).

**Figure 3 plants-14-00654-f003:**
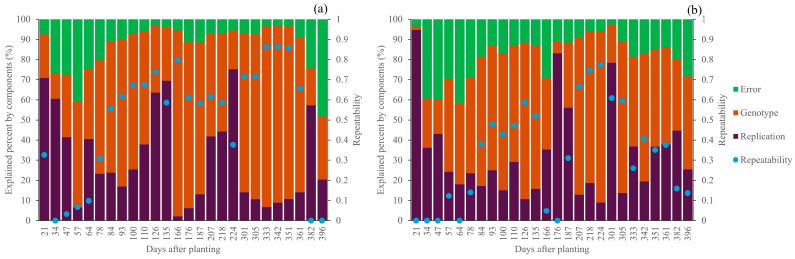
Variance component decomposition for temporal canopy cover (**a**) and canopy height (**b**).

**Figure 4 plants-14-00654-f004:**
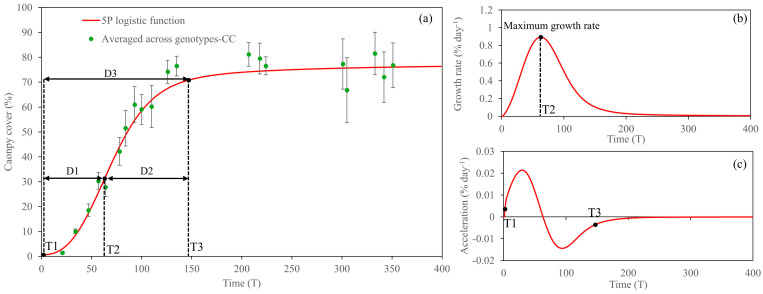
Fitted five-parameter logistic growth over temporal canopy cover (CC) (**a**), its first derivative (**b**), and second derivative (**c**). T and D refer to time and duration, respectively.

**Figure 5 plants-14-00654-f005:**
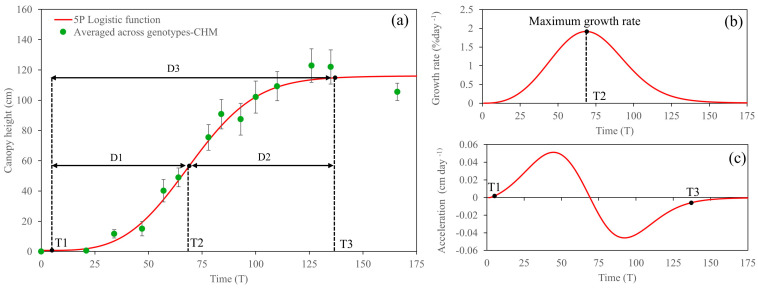
Fitted five-parameter logistic growth over temporal canopy height before cold burn (**a**), its first derivative (**b**), and second derivative (**c**). T and D refer to time and duration, respectively.

**Figure 6 plants-14-00654-f006:**
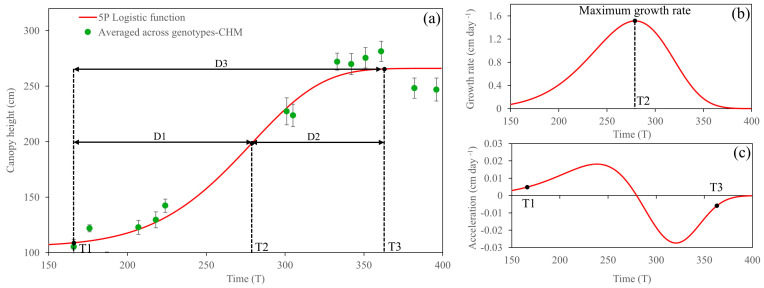
Fitted five-parameter logistic growth analysis over temporal canopy height after cold burn (**a**), its first derivative (**b**), and second derivative (**c**). T and D refer to time and duration, respectively.

**Figure 7 plants-14-00654-f007:**
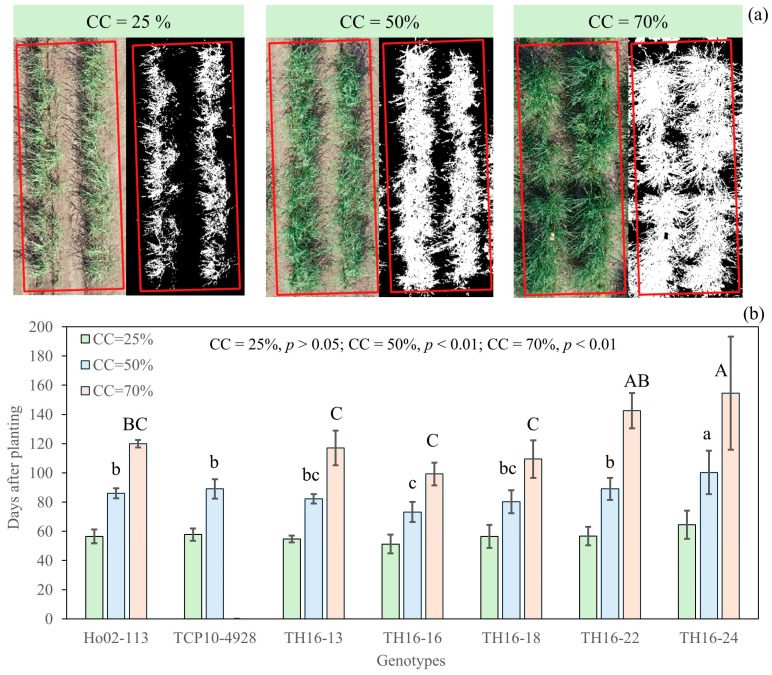
Red, green, and blue (RGB) orthomosaic and canopy cover for 25%, 50%, and 70% (**a**). The number of days it took to achieve 25%, 50%, and 70% canopy cover for seven energy cane genotypes (**b**). Lowercase and uppercase letters are LSD analysis for 50 and 70%, respectively.

**Figure 8 plants-14-00654-f008:**
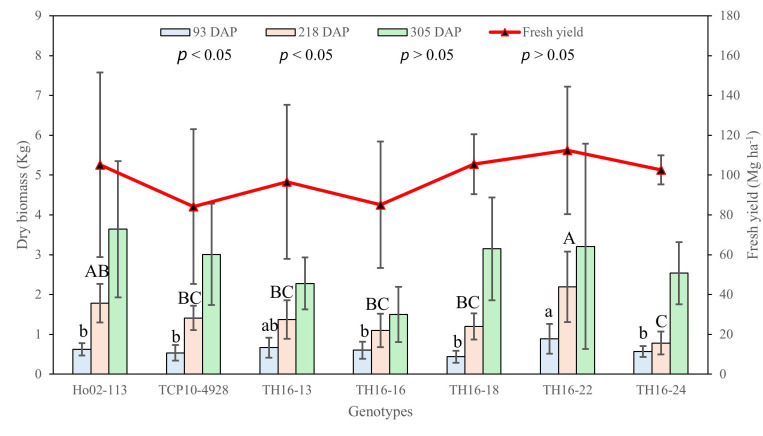
Dry biomass of seven energy cane genotypes at 93, 218, and 305 days after planting (DAP) and fresh yield. Bar graphs with the same letters are not significantly different by LSD at *p* < 0.05.

**Figure 9 plants-14-00654-f009:**
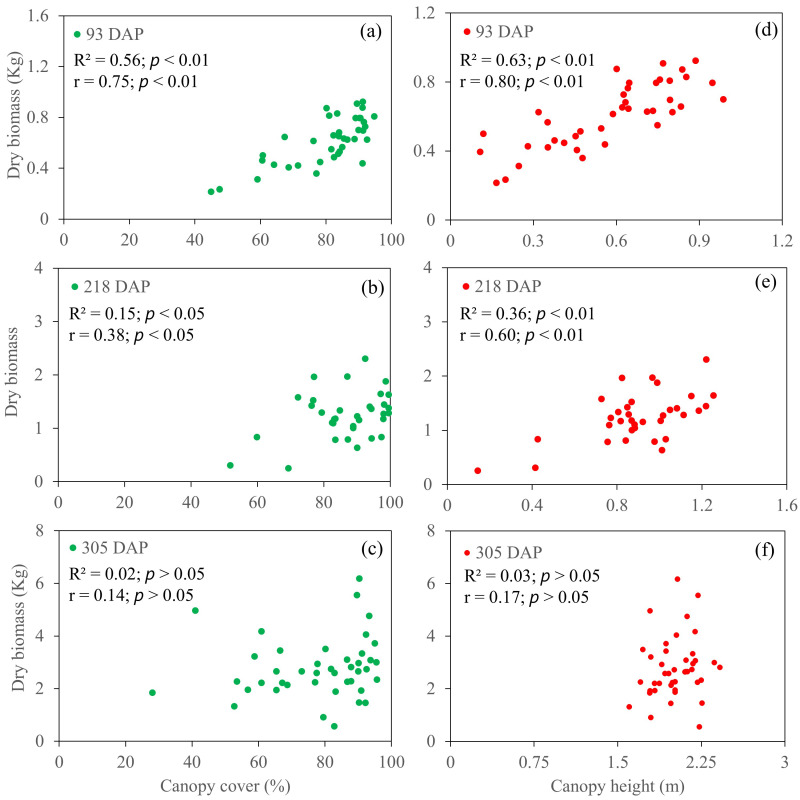
Relationship between uncrewed aerial system (UAS)-based canopy cover (**a**–**c**) and canopy height (**d**–**f**) and dry biomass.

**Figure 10 plants-14-00654-f010:**
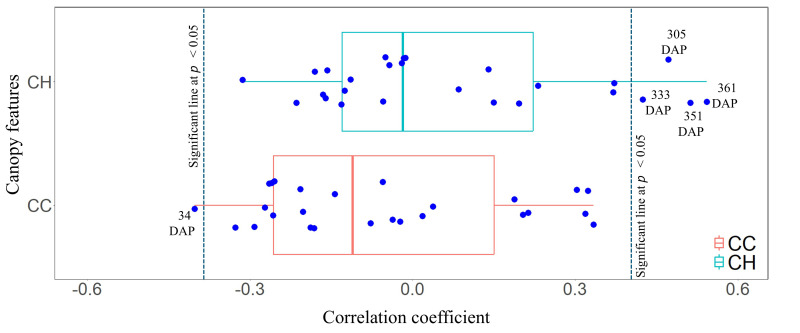
Boxplot of correlation coefficients between fresh yield of energy cane and canopy cover (CC) and canopy height (CH) measurements collected across the growing season. The abbreviation DAP refers to days after planting.

**Figure 11 plants-14-00654-f011:**
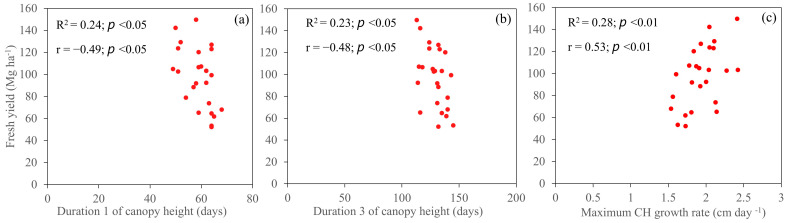
Relationship between fresh yield of energy cane and canopy parameters. Duration 1 (**a**), duration 3 (**b**), and maximum CH growth rate (**c**) of canopy height before cold burn.

**Figure 12 plants-14-00654-f012:**
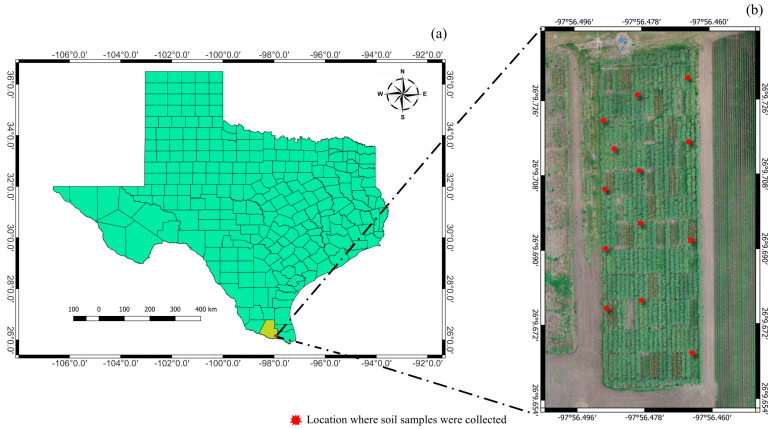
Study area in this work, located in Weslaco, Texas. A map showing the whole state with Hidalgo County highlighted in yellow (**a**), and the trial in this study (**b**).

**Figure 13 plants-14-00654-f013:**
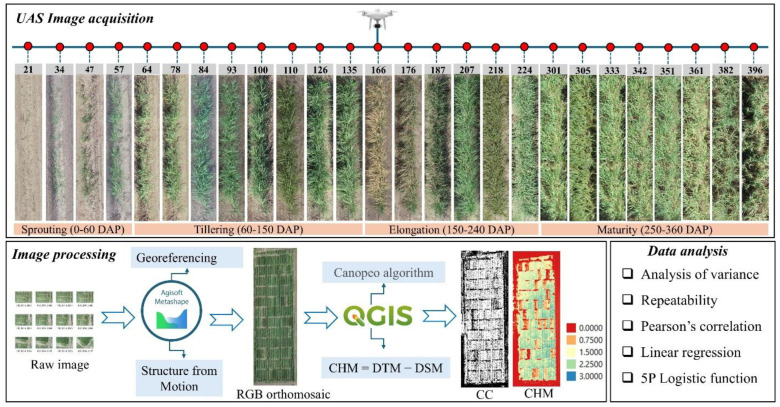
A flowchart of using uncrewed aerial system (UAS)-based high-throughput phenotyping for growth analysis. The numbers in the gray boxes indicate days after planting (DAP).

**Table 1 plants-14-00654-t001:** Canopy cover growth parameters derived from the five-parameter logistic function, including duration 1 (D1), duration 2 (D2), duration 3 (D3), and the maximum growth rate (Max. rate).

Genotypes	Canopy Cover Growth Parameters			
	D1 (Days)	D2 (Days)	D3 (Days)	Max. Rate (% Day^−1^) (T2)
Ho02-113	64.50 ± 5.19	86.75 ± 3.77	151.25 ± 7.41	0.89 ± 0.05
TCP10-4928	60.50 ± 4.20	68.25 ± 7.41	128.75 ± 11.32	0.87 ± 0.06
TH16-13	60.75 ± 4.27	77.50 ± 9.32	138.25 ± 7.71	0.96 ± 0.01
TH16-16	56.00 ± 9.56	73.75 ± 5.50	129.75 ± 5.43	1.19 ± 0.05
TH16-18	59.00 ± 4.24	74.25 ± 10.07	133.25 ± 8.53	1.10 ± 0.16
TH16-22	60.00 ± 12.11	85.75 ± 2.75	145.75 ± 14.86	0.84 ± 0.07
TH16-24	59.75 ± 8.05	66.75 ± 10.93	126.50 ± 13.86	0.82 ± 0.14
F-test	ns	*	*	**
LSD_0.05_	−	12.29	16.02	0.14

ns, *, and ** indicate non-significant, and significant differences at 0.05 and 0.01, respectively.

**Table 2 plants-14-00654-t002:** Canopy height growth parameters before cold burn derived from the five-parameter logistic function, including duration 1 (D1), D2, D3, and the maximum growth rate (Max. rate).

Genotypes	Canopy Height Growth Parameters			
	D1 (Days)	D2 (Days)	D3 (Days)	Max. Rate (cm Day^−1^) (T2)
Ho02-113	59.50 ± 9.14	77.0 ± 8.60	136.50 ± 13.86	1.66 ± 0.26
TCP10-4928	66.25 ± 2.06	71.0 ± 2.94	137.25 ± 2.63	1.64 ± 0.13
TH16-13	50.75 ± 5.79	72.50 ± 2.08	123.25 ± 7.80	2.17 ± 0.29
TH16-16	58.25 ± 5.37	70.25 ± 10.04	128.50 ± 9.29	1.99 ± 0.25
TH16-18	63.00 ± 1.15	68.00 ± 11.57	131.00 ± 12.24	1.89 ± 0.19
TH16-22	61.75 ± 2.63	64.75 ± 6.55	126.50 ± 9.14	2.26 ± 0.17
TH16-24	56.50 ± 5.06	66.25 ± 10.99	122.75 ± 7.50	1.84 ± 0.05
F-test	**	ns	ns	**
LSD_0.05_	6.89	−	−	0.29

ns and ** indicates not significant and significant difference at 0.01%, respectively.

**Table 3 plants-14-00654-t003:** Canopy height growth parameters after cold burn derived from the five-parameter logistic function, including duration 1 (D1), D2, D3, and the maximum growth rate (Max. rate).

Genotypes	Canopy Height Growth Parameters			
	D1 (Days)	D2 (Days)	D3 (Days)	Max. Rate (cm Day^−1^) (T2)
Ho02-113	121.50 ± 11.26	102.00 ± 15.44	223.50 ± 26.66	1.27 ± 0.05
TCP10-4928	112.00 ± 16.67	86.00 ± 22.07	198.00 ± 38.58	1.50 ± 0.31
TH16-13	106.00 ± 4.89	76.25 ± 4.34	182.25 ± 9.10	1.62 ± 0.17
TH16-16	108.50 ± 11.35	77.75 ± 14.97	186.25 ± 26.27	1.75 ± 0.34
TH16-18	108.00 ± 8.71	77.00 ± 13.11	185.00 ± 21.78	1.80 ± 0.27
TH16-22	105.25 ± 9.70	74.75 ± 12.23	180.00 ± 21.83	1.73 ± 0.18
TH16-24	116.50 ± 5.80	88.00 ± 7.39	204.50 ± 13.17	1.56 ± 0.09
F-test	ns	ns	ns	ns
LSD_0.05_	−	−	−	−

ns indicates non-significant difference at 0.05%.

**Table 4 plants-14-00654-t004:** Soil fertility analysis for the trial.

Parameters	Value	Unit
PH	8.00	−
Conductivity	551.00	umho/cm
Nitrate	41.33	mg/kg
Phosphorus	31.67	mg/kg
Potassium	443.00	mg/kg
Calcium	9566.33	mg/kg
Magnesium	603.67	mg/kg
Sulfur	119.33	mg/kg

**Table 5 plants-14-00654-t005:** Summary of biomass and fresh yield data collection for the energy cane.

Ground Measurement	Phenological Stage	Days After Planting	Methods
Dry biomass	Tillering	93	Manual harvesting
Dry biomass	Elongation	218	Manual harvesting
Dry biomass	Early maturity	305	Manual harvesting
Fresh yield (final yield)	Late maturity	460	Mechanical harvesting

## Data Availability

The raw data supporting the conclusions of this article will be made available by the authors on request.

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
