# Peer review of "Use of Uncrewed Aerial System (UAS)-Based Crop Features to Perform Growth Analysis of Energy Cane Genotypes"

_plants, 2025, doi:10.3390/plants14050654_

Round 1
Reviewer 1 Report
Comments and Suggestions for Authors
The authors presented a remarkable and very detailed study. However, I believe that the article will be better if the authors add a number of additions.
1. The authors of the article do not describe in any way the procedure for identifying the green mass in the images and the magnitude of the errors of the method used.
2. The authors provide data on the mineral content in the soil in Table 4. However, they did not indicate how these figures were obtained. Is it an averaging over multiple measurement points or is it the result of a single measurement? If this is the result of a measurement at a single point, this is clearly not enough because the mineral content can vary significantly within even a small field. To obtain the result correctly, it is advisable to perform such an analysis at each point corresponding to the zones in Fig.1.
Reviewer 2 Report
Comments and Suggestions for Authors
The manuscript utilizes an Unmanned Aerial System (UAS) to continuously acquire phenotypic data of different sugarcane varieties and analyzes their growth conditions. The content of the article is relatively comprehensive, and the analysis is thorough. However, the following issues remain to be discussed:
(1) When calculating CC, the authors employed a color ratio method, which can be influenced by varying lighting conditions. Has this aspect been considered in the paper, and how was it addressed?
(2) In section 4.3, data was collected at 93, 128, and 305 Days After Planting (DAP). Were the phenological stages of each variety consistent at these time points? Did this have any impact on the experimental results?
(3) Why did the authors choose the 5-parameter logistic (5PL) function for fitting?
Reviewer 3 Report
Comments and Suggestions for Authors
Use of Uncrewed Aerial Systems (UAS) Based – Crop Features to Perform Growth Analysis of Energy Cane Genotypes
1. Very interesting research entitled “Use of Uncrewed Aerial Systems (UAS) Based – Crop Features to Perform Growth Analysis of Energy Cane Genotypes”.
2. Correct the structure of the article. Adapt it to the structure of plants-MDPI (see attached file).
3. The sections marked in yellow in point 2 are incorrectly numbered. (3.3, 3.3.1 and 3.3.2 – lines: 262, 264 and 280). Correct.
4. Delete the sections: “Institutional Review Board Statement” and “Informed Consent Statement” are not used in Plants-MDPI.
5. Check Microsoft Word template at: https://www.mdpi.com/files/word-templates/plants-template.dot
6. The title of figures 1, 2, 3, 4, 5, 6, 7, 8, 9 10 and 11 is too long. The title should be short (maximum two lines), explaining the figure in the previous paragraph.
7. The title of tables 1, 2 and 3 is too long. The title should be one or two lines (maximum).
8. I suggest that the images be made a little larger, so that they can be seen better and the text can be read.
9. After taking the images of the Energy Cane crop (lines 468-477), some preprocessing was done to correct (noise, too much light, too little light, shadows, etc.). Explain.
10. In what color model are the images obtained (RGB, rgb, XYZ, L*a*b*, L*u*v*, HSV, HLS, YCrCb, YUV, I1I2I3, TSL, etc) for processing. Is there any conversion?
11. I suggest showing the “Canopeo algorithm” used in your research. Consider the following steps:
- Image Acquisition
- Preprocessing (cropping, removing too much light, removing shadows, etc.)
- Color model change, if applicable (RGB, rgb, XYZ, L*a*b*, L*u*v*, HSV, HLS, YCrCb, YUV, I1I2I3, TSL, etc.).
- Segmentation
- Feature Extraction
- Classification or recognition of Energy Cane
- Results
12. I suggest that the algorithm in this article use the following format: (see attached file).
13. Consider future work on this research.
14. Very good bibliography.
The article has good content and very interesting.
Authors are requested to make all indicated corrections.

English language proofreading by a native English speaker is required.
Round 2
Reviewer 2 Report
Comments and Suggestions for Authors
My conerns have been addressed. Recommend for acceptance.
Reviewer 3 Report
Comments and Suggestions for Authors
I thank the authors for having made the indicated corrections and improvements to the article in general.
Comments on the Quality of English LanguageNo comments.